# Modeling the Price Volatility of Cassava Chips in Thailand: Evidence from Bayesian GARCH-X Estimates

Jittima Singvejsakul *, Yaovarate Chaovanapoonphol and Budsara Limnirankul

Department of Agricultural Economy and Development, Faculty of Agriculture, Chiang Mai University, Chiang Mai 50200, Thailand; yaovarate.chao@cmu.ac.th (Y.C.); budsara.l@cmu.ac.th (B.L.)
* Correspondence: jittima.s@cmu.ac.th

**Abstract:** Thailand is a significant global exporter of cassava, of which cassava chips are the main export products. Moreover, China was the most important export market for Thailand from 2000 to 2020. However, during that period, Thailand confronted fluctuations in the cassava product price, and cassava chips were a product with significant price volatility, adapting to changes in export volumes. This study aims to analyze the volatility of the price of cassava chips in Thailand from 2010 to 2020. The data were collected monthly from 2010 to 2020, including the price of cassava chips in Thailand (Y), the volume of cassava China imported from Thailand (X1), the price of the cassava chips that China imported from Thailand (X2), the price of the cassava starch that China imported from Thailand (X3), the substitute crop price for maize (X4), the substitute crop price for wheat (X5), and Thailand's cassava product export volume (X6). The volatility and the factors affecting the volatility in the price of cassava chips were calculated using Bayesian GARCH-X. The results indicate that the increase in X1, X2, X3, X4, and X6 led to an increase in the rate of change in cassava chip price volatility. On the other hand, if the substitute crop price for wheat (X5) increases, then the rate of change in the volatility of the cassava chip price decreases. Therefore, the government's formulation of an appropriate cassava policy should take volatility and the factors affecting price volatility into account. Additionally, the government's formulation of agricultural policy needs to consider Thailand's macro-environmental factors and its key trading partners, especially when these environmental factors signal changes in the price volatility of cassava.

**Keywords:** cassava price; volatility; Bayesian; GARCH-X; Thailand

## 1. Introduction

Thailand is a significant global exporter of cassava, and to meet increasing world demand for cassava, Thailand's cassava plantation area and yield have increased from 7,400,148 million hectares (yield 21,912,416 million tons) in 2010, to 9,319,718 million hectares (yield 32,357,741 million tons) and 9,439,009 million hectares (yield 28,999,122 million tons) in 2015 and 2020, respectively (Office of Agricultural Economics 2020). Cassava is an inexpensive crop compared to other starchy food crops, and is used as a raw material in the food industry, animal feed, bio-energy, and industries such as alcohol, citric acid, clothing, and chemicals. Consequently, the demand for cassava in the world market has continued to increase, and it has become an essential global economic crop after wheat, corn, rice, and potatoes. Meanwhile, global cassava production has also continued to increase. Between 2010 and 2019, global cassava output increased from 242.07 million tons to 277.07 million tons (FAO 2020).

Thailand's cassava products are processed into two product types, dried cassava (cassava chips, cassava pellets, and other types of cassava) and primary cassava products (native starch and modified starch). A total of 64% of cassava produced in Thailand is exported as different types of products (Office of Agricultural Economics 2020). In 2010, Thailand exported 4,611,976 million tons of cassava chips and 2,235,574 million tons of native starch. Through to 2020, Thailand's cassava product export structure remained the

same, with cassava chips being the most exported (3,063,671 million tons), followed by native starch (2,781,681 million tons) (Department of Business Development, Ministry of Commerce 2013). Thailand is still the world's largest exporter of cassava products, and European and Asian markets represent the main important export markets (Thai Tapioca Starch Association 2020). Yet the quantity of cassava exports to the European market has tended to decrease due to policies aiming to reduce imports of cassava pellets and substitutes to the EU market. The Asian market has thus become Thailand's primary export market for cassava products, replacing the EU market. China has been Thailand's most crucial cassava product export market since 2000, with 60% of Thailand's cassava product export volume being exported to China for use in the animal industry and for the production of alcohol, citric acid, and ethanol (Ministry of Commerce 2020).

The expansion of Thailand's cassava industry in the past ten years resulted in the expansion of the export market, especially the Chinese market, due to rising demand for cassava products in the animal industry. Additionally, from 2012 to 2014 the Chinese government supported ethanol production to develop biofuels and reduce dependence on oil imports. As a result, the demand for cassava chips for ethanol production increased significantly, since cassava prices were lower than other starchy food crops. However, in 2018, China introduced a policy to support maize instead of cassava in the ethanol production industry, resulting in a decline of the feed industry. Nevertheless, the prices of other starchy food crops declined and cassava demand in China decreased. This affected Thailand's cassava export volume which shrank by 26.1% in 2018 and 20.7% in 2019. The price of cassava in Thailand subsequently dropped sharply. Considering the price volatility of cassava products in Thailand between 2000 and 2020, cassava chips were clearly a product with significant price volatility, adapting to changes in export volumes during that period (The World Bank 2020).

The price volatility of cassava chip products in Thailand, especially the cassava chips that are the main export products, affect farmers' cassava planting plans, including setting up government policies to ensure farmers' income. Empirical evidence of what causes the price volatility of cassava products in Thailand will help improve the accuracy of policies developed by organisations. In the past, the characteristics of price volatility of agricultural products in Thailand and elsewhere has been widely studied using the GARCH models, both the symmetry model (e.g., GARCH [1,1]) and the asymmetry model (e.g., EGARCH, TGARCH and PGARCH), such as the rice price volatility by the generalized autoregressive conditional heteroscedastic (GARCH) method (Baharom et al. 2009). These resulted in findings of the relationship between purchasing volume and yield in the agricultural futures market using the GARCH Model (Boonvorachote and Thongsit 2013). The estimation of economic return and price volatility of agricultural exports in Indonesia (Hatane 2011) has been studied using both traditional symmetric (GARCH) and asymmetric models (EGARCH, TGACRH). The study found that analytical results from the symmetric GARCH model were the best estimation of parameters, which had similar results to price volatility of agricultural product found by Beck (2001), O'Connor et al. (2009), and Mahesha (2011), which also applied GARCH to analyze price volatility in agricultural products.

These models only consider the price volatility of agricultural products without considering other factors that affect price volatility. This is inconsistent with empirical evidence where the natural nature of prices does not cause fluctuations in agricultural prices, but the price volatility of each agricultural product is often affected by both domestic and foreign factors. The GARCH-X model is used to analyze the impact of factors affecting price fluctuations, such as the food price volatility in Greece resulting from macro-economic factors (Apergis and Rezitis 2011) using both the GARCH and GARCH-X models, and the price volatility of agricultural commodities in China affected by macro-economic factors (Xue and Sriboonchitta 2014). With the above empirical evidence, changes in the price of cassava products in Thailand are likely to be directly affected by Chinese government policies and the substitution of other starchy food crop prices. This study uses the GARCH-X model to analyze the impact of factors on the volatility of cassava products in Thailand.

Although the model considers the factors that cause price volatility, parameter estimation uses the maximum likelihood method to obtain fixed parameters, which are estimations obtained only through optimization. This study adjusts the estimations of parameters by adopting Bayesian estimation, which assumes that parameters are random variables that have a probability distribution. This provides a more accurate result where an estimator is closer to the true value, in particular, where volatility results are affected by a factor that is difficult to predict. Furthermore, Bayesian estimation is an excellent tool to estimate the parameters of the model and examine the uncertainty of estimation for appropriate affects and accurate analysis. Moreover, the model also benefits farmers and governments concerned with production planning and policy issuance to resolve problems arising from agricultural price fluctuations.

Previous studies of agricultural price volatility show continued modelling development, starting with a fundamental model such as the GARCH Model. The GARCH model has been used to analyze agricultural price volatility over various periods and situations, such as Yang et al. (2010) which studied the volatility of agricultural products due to free price policy or liberalization policies in the United States. The study found that the adoption of free trade policies in agriculture increased price volatility for three main agricultural commodities, corn, soybeans, and wheat, while it decreased the volatility of cotton fiber. The same applies to Sendhil et al. (2014) which studied the price volatility of maize, soybean, cottonseed, oilcake, castor, palm oil, cumin, and chili pepper in India. The results of that study found that having future markets in the country helped to reduce agricultural price volatility. Moreover, Čermák (2017) studied volatility through agricultural commodity market modelling in the Czech Republic (especially the price of agricultural commodities such as wheat and maize) in the period of global economic crisis in 2008. Čermák observed price volatility of corn and wheat. While the GARCH model has been used to study the price volatility of agricultural commodities continuously until the present, this model has limitations in that it does not consider factors related to volatility, and it only estimates the parameters of variables in fluctuation characteristics (Hwang and Satchell 2001).

The GARCH-X Model by Braun et al. (1995) is a model used to analyze agricultural price volatility since it considers external key factors that affect the volatility that occurs. For example, Apergis and Rezitis (2011) used the GARCH and GARCH-X models to analyze food price fluctuations affected by short-term changes in food prices and macroeconomic factors in Greece. The estimated results from the GARCH-X model showed a positive impact on food price volatility when the government intervened in allocating domestic resources. In addition, the GARCH-X model provides better estimation results than the GARCH model. Xue and Sriboonchitta (2014) studied how the volatility of China's agricultural price index was affected by macro-economic factors such as domestic price factors. The study indicated that domestic prices factors have important short-term effects on the price volatility of agricultural products in China. Moreover, the study compared the estimation efficacy of the GARCH-X model with the EGARCH, GJR-GARCH, and GARCH-t models. The results indicated that the GARCH-X model had the lowest value of AIC BIC (selection model). There was a cointegration relationship between macro-economic factors and domestic prices.

Although the GARCH-X model takes external factors into account as a fix for the GARCH model, the estimation of the GARCH-X model is based on the maximum likelihood estimation method, which has disadvantages. The maximum likelihood estimation is an optimization estimation which has a complex and sensitive approach to initial optimization, and the estimation can be inaccurate if the sample size is small (Virginia et al. 2020). Therefore, this study applies Bayesian estimation in GARCH-X modeling to obtain a qualified estimator, using an MCMC algorithm for sampling estimation to find parameters can approach true value even small sample size. In addition, parameters characteristics are random parameters, unlike the maximum likelihood estimation, which provides fixed parameters. This price volatility analysis of cassava chips uses the GARCH-X model through the Bayesian estimation method. The volatility model analysis is developed to

correct GARCH-X model deficiencies and obtain more accurate parameters that are not currently found in Thai agricultural commodity price volatility studies

## 2. Research Methodology

### 2.1. The Unit Root Test Using Bayesian Estimation

The unit root test was investigated using the ADF test, which shows the ratio between stationary data and the non-stationary data of the null hypothesis (Dickey and Said 1981). The significant statistical issues associated with the autoregressive unit root test (AR) are defined as

$$x_t = c + \rho x_{t-1} + \varepsilon_t, \ \varepsilon_t \sim N(0, \sigma^2), \tag{1}$$

The prior density of $\rho$ is formulated and expressed as the following:

$$p(\theta) = p(\phi)p(a^*|\phi), \tag{2}$$

The marginal likelihood for $\phi$ is

$$l(\phi|D)\alpha \int l(\rho|D)\phi(a^*|\phi)da^*, \tag{3}$$

The consideration of the hypotheses of Bayesian estimation was combined with the Bayes factor to interpret the hypothesis of stationary data. The null hypothesis is defined by $N_i$ and the alternative hypothesis is denoted by $N_j$. The ratio of the posterior odds of $N_i$ and $N_j$ is

$$\frac{p(M_i|y)}{p(M_j|y)} = \frac{p(y|M_i)}{p(y|M_j)} \times \frac{\pi(M_i)}{M_j}, \tag{4}$$

The Bayes factor can be interpreted in Table 1.

**Table 1.** The implication of Bayes factor of the Jeffrey guideline model.

| Items | Interpretation |
|---|---|
| BF < 1/10 | Strong evidence for $M_j$ |
| 1/10 < BF < 1/3 | Moderate evidence for $M_j$ |
| 1/3 < BF < 1 | Weak evidence for $M_j$ |
| 1 < BF < 3 | Weak evidence for $M_i$ |
| 3 < BF < 10 | Moderate evidence for $M_i$ |
| 10 < BF | Strong evidence for $M_i$ |

Source: modified from Jeffreys (1946) by authors.

### 2.2. The GARCH-X Using Bayesian Estimation

To model the volatility of the agricultural sector, especially in commodity prices, the autoregressive conditional heteroscedasticity (ARCH) of Engle (1982) and the generalized ARCH (GARCH) of Bollerslev (1982) are obvious ways to measure volatility. To understand the GARCH-X model, first, we introduce the GARCH model, which can be written as

$$h_t = \omega + \alpha\varepsilon_t^2 + \beta h_{t-1}, \tag{5}$$

where the variance of $\varepsilon_t^2$ is 1 and also this only works if $\alpha + \beta < 1$ and $\alpha > 0$, $\beta > 0$, $\omega > 0$. The GARCH-X model aimed to model the conditional variance. Then, the specification equation of GARCH model can be expressed as

$$h_t = \omega + \alpha\varepsilon_t^2 + \beta h_{t-1} + \gamma x, \tag{6}$$

where $\omega > 0$, $\alpha_1, \beta \geq 0$, the conditional variance is finite, and the restrictions on the GARCH parameters $\alpha_0$, $\alpha_1$ and $\beta$ guarantee its positivity.

In order to write the likelihood function, we define the vectors $\varpi = (\varpi_1, \ldots, \varpi_T)$, $\alpha = (\alpha_0, \ldots, \alpha_t)$, $\beta = (\beta_0, \ldots, \beta_t)$, and $\gamma = (\gamma_0, \ldots, \gamma_t)$. We regroup the model parameters into the vector $\psi = (\omega, \alpha, \beta, \gamma)$. Then, upon defining the $T \times T$ diagonal matrix

$$\sum = \sum(\psi,) = diag\left(\{h_t(\omega, \alpha, \beta, \gamma)\}_{t-1}\right), \tag{7}$$

where $h_t(\alpha, \beta) = \alpha_0 + \alpha_1 y_{t-1}^2 + \beta h_{t-1}(\alpha, \beta)$, we can express the likelihood of $(\psi)$ as

$$L(\psi|y)\alpha\left(\det \sum\right)^{-1/2} \exp\left[-\frac{1}{2}y'\overset{-1}{\sum}y\right], \tag{8}$$

The Bayesian approach considers $(\psi)$ as a random variable that is characterized by a prior density denoted by $p(\psi)$. The prior is specified with the help of parameters called hyperparameters which are initially assumed to be known and constant. Moreover, depending on the researcher's prior information, this density can be more or less informative. Then, by coupling the likelihood function of the model parameters with the prior density, we can transform the probability density using Bayes' rule to get the posterior density $p(\psi|\,|y)$ as follows:

$$p(\psi|y) = \frac{L(\psi,|y)p(\psi)}{\int L(\psi|y)p(\psi)d\psi}, \tag{9}$$

This posterior is a quantitative, probabilistic description of the knowledge about the model parameters after observing the data. For an excellent introduction on Bayesian econometrics, we refer the reader to Koop (2003).

The joint prior distribution is then formed by assuming prior independence between the parameters, i.e.,

$$p(\psi) = p(a)p(\beta). \tag{10}$$

The recursive nature the GARCH-X variance equation implies cannot be expressed in closed form. There exists no (conjugate) prior that can remedy this property. Therefore, we cannot use the simple Gibbs sampler and need to rely on a more elaborated Markov Chain Monte Carlo (MCMC) simulation strategy to approximate the posterior density. The idea of MCMC sampling was first introduced by Metropolis et al. (1953) and was subsequently generalized by Hastings (1970). The sampling strategy relies on the construction of a Markov chain with realizations $\left(\psi^{[0]}\right), \ldots, \left(\psi^{[j]}\right), \ldots$ in the parameter space. Under appropriate regularity conditions, asymptotic results guarantee that as $j$ tends to infinity, $\left(\psi^{[j]}\right)$ tends in distribution to a random variable. Hence, after discarding a burn-in of the first draws, the realized values of the chain can be used to make inferences about the joint posterior.

### 3. Empirical Results

#### 3.1. Data Descriptive

The cassava price data and the other variables that were considered in this study consist of seven sets of agricultural data, including cassava chips prices in Thailand (y), China's cassava import volume from Thailand (X1), China's cassava chips import price from Thailand (X2), China's cassava starch import price from Thailand (X3), the substitute crop price of maize (X4), the substitute crop price of wheat (X5), Thailand's cassava products export volume (X6). All of the data was monthly data in 105 observations from 2010–2020. Basically, the basic information consists of a mean value, maximum and minimum values, standard deviation, skewness, kurtosis, Jarque–Bera, probability, summation and observations in Table 2.

**Table 2.** Summary statistics of key variables in Bayesian GARCH-X Model.

| Statistics | Y | X1 | X2 | X3 | X4 | X5 | X6 |
|---|---|---|---|---|---|---|---|
| Mean | 0.0019 | 0.0042 | 0.0016 | 0.0031 | 0.0019 | 0.0028 | 0.0019 |
| Median | 0.0110 | −0.0172 | 0.0000 | −0.0093 | 0.0010 | 0.0000 | −0.0317 |
| Max. | 0.2943 | 3.4250 | 0.2877 | 3.3891 | 0.3236 | 0.2395 | 3.1231 |
| Min. | −0.3044 | −2.5720 | −0.2881 | −2.5388 | −0.3332 | −0.2332 | −2.2706 |
| Std.Dev. | 0.1931 | 0.6841 | 0.1894 | 0.6777 | 0.1956 | 0.1886 | 0.6254 |
| Skewness | −0.0578 | 0.7161 | −0.0364 | 0.6982 | −0.0516 | −0.0121 | 0.7895 |
| Kurtosis | 1.5083 | 9.3457 | 1.5001 | 9.3080 | 1.5821 | 1.4137 | 9.1423 |
| Jarque-Bera | 9.6073 | 181.6222 | 9.6779 | 179.1358 | 8.6737 | 10.8018 | 172.6151 |
| Probability | 0.0082 | 0.0000 | 0.0079 | 0.0000 | 0.0131 | 0.0045 | 0.0000 |

Source: authors' estimation.

## 3.2. Stationary Testing

Empirically, all variables are included in the model are time series data. Therefore, the data should be tested to determine if it is stationary. In this paper, the unit root test based on the Bayesian method was used to investigate the stationary data, which is shown in Table 3. The null hypothesis ($H_0$) is non-stationary and the alternative hypothesis ($H_1$) is stationary. The results show that all variables are stationary or ($I(0)$).

**Table 3.** Unit root testing of key variables relies on the Bayesian inference.

| Variables | Bayesian Factor Ratios (M1/M2) | Implication | Result |
|---|---|---|---|
| Cassava chip price of Thailand (Y) | $1.65 \times 10^{-31}$ | Strong evidence for Mj | I(0) |
| China's cassava import volume from Thailand (X1) | $2.2 \times 10^{-17}$ | Strong evidence for Mj | I(0) |
| China's cassava chips import price from Thailand (X2) | $5.3 \times 10^{-32}$ | Strong evidence for Mj | I(0) |
| China's cassava starch import price from Thailand (X3) | $1.14 \times 10^{-17}$ | Strong evidence for Mj | I(0) |
| Substitute crop price: maize (X4) | $3.33 \times 10^{-29}$ | Strong evidence for Mj | I(0) |
| Substitute crop price: wheat (X5) | $3.97 \times 10^{-39}$ | Strong evidence for Mj | I(0) |
| Thailand's cassava products export volume (X6) | $2.24 \times 10^{-16}$ | Strong evidence for Mj | I(0) |

Source: authors' estimation.

## 3.3. The Estimation of GARCH-X(1,1) Using Bayesian Inference

In this study, we applied the GARCH-X model using Bayesian estimation to investigate the volatility of the cassava chip price, which is influenced by the factors. This study applies Bayesian estimation using the MCMC method because the estimated parameters from the MCMC method iares the most effortless procedure to obtain the posterior of the PDF condition, involving the computation of the expected value for the properties on the GARCH-X model. The number of iterations of the Markov chain sampling was 10,000, and we identified the first 2000 as burn-in; thus, the size of the Monte Carlo is 8000. The posterior means and the standard deviations are presented in Tables 4 and 5.

The empirical results show that for the ARCH term ($\alpha$), all the six variables were tightly distributed around the value of unity, indicating that they follow the random walk. The posterior means and standard deviation show that the ARCH term ($\alpha$) is statistically significant within the invertible region. In comparison, the intercept terms ($\omega$) are extremely small for all of the six variables. For the properties of the GARCH term ($\beta$), the coefficients indicate exogenous factors influence the volatility of cassava chip price. The variable of China's cassava import volume from Thailand (X1) shows that the $\beta$ coefficient is 0.0326, which is within the credible interval that is statistically significant. Moreover, the coefficients of the other variables including China's cassava chips import price from Thailand (X2), China's cassava starch import price from Thailand (X3), substitute crop price: maize (X4), substitute crop price: wheat (X5), and Thailand's cassava products export

volume (X6) were 0.0234, 0.3618, 0.0249, 0.0240, and 0.0233, respectively. The volatility reacts the ARCH term, and the GARCH term provides the persistence of overall volatility. Furthermore, the X term ($\gamma$) for most exogenous factors except the substitute crop price of wheat (X5), shows positive impact on the volatility of the cassava chip price. This indicates that increases in X1, X2, X3, X4, and X6 lead to increases of the change of volatility of the cassava chip price. On the other hand, if the substitute crop price of wheat (X5) increases, then it causes a decrease of the change of the volatility of the cassava chip price. Moreover, the posterior means of all the variables are also tightly within the credible interval value, indicating that they are statistically significant.

**Table 4.** The posterior means, standard deviation and credible intervals of the parameters.

| Variables | X1 | | X2 | | X3 | |
|---|---|---|---|---|---|---|
| | Coefficient | 95%CI | Coefficient | 95%CI | Coefficient | 95%CI |
| $\omega$ | −0.0002 (0.000473) | (−0.0006, 0.0012) | 0.0002 (0.0004) | (−0.0005, 0.0012) | 0.0003 (0.0040) | (−0.0006, 0.0210) |
| $\alpha$ | 0.0023 (0.100100) | (0.0021, 0.1927) | 0.0025 (0.0990) | (0.0021, 0.1922) | 0.0024 (0.0091) | (0.0021, 0.1920) |
| $\beta$ | 0.0326 (0.090000) | (0.0100, 0.6900) | 0.0234 (0.0900) | (0.0960, 0.7100) | 0.3618 (0.0910) | (0.0396, 0.9071) |
| $\gamma$ | 0.0006 (0.000494) | (0.0001, 0.0009) | 0.0010 (0.0045) | (0.0009, 0.0015) | 0.0001 (0.0005) | (0.00009, 0.0096) |
| Sigma2 | 0.0001 (0.000002) | (0.00009, 0.00024) | 0.0002 (0.0010) | (0.00008, 0.0010) | 0.0035 (0.0001) | (0.0006, 0.0140) |

Note: figures in the parentheses are standard deviations. Source: authors' estimation.

**Table 5.** The posterior means, standard deviation and credible intervals of the parameters.

| Variables | X4 | | X5 | | X6 | |
|---|---|---|---|---|---|---|
| | Coefficient | 95%CI | Coefficient | 95%CI | Coefficient | 95%CI |
| $\omega$ | −0.0029 (0.0005) | (−0.0065, 0.0013) | 0.0037 (0.0050) | (−0.0067, 0.0062) | −0.0062 (0.0054) | (−0.0560, 0.0122) |
| $\alpha$ | 0.0032 (0.0104) | (0.0020, 0.1935) | 0.0025 (0.1004) | (0.0020, 0.1931) | 0.0029 (0.0900) | (0.0019, 0.0640) |
| $\beta$ | 0.0249 (0.0800) | (0.0003, 0.7690) | 0.0240 (0.0800) | (0.0090, 0.7000) | 0.0233 (0.9990) | (0.0037, 0.9071) |
| $\gamma$ | 0.0600 (0.0019) | (0.0003, 0.0786) | −0.0210 (0.0207) | (−0.0411, 0.0030() | 0.0102 (0.0006) | (0.0090, 0.0102) |
| Sigma2 | 0.0010 (0.0002) | (0.0076, 0.0035) | 0.0100 (0.0100) | (0.00008, 0.0014) | 0.0004 (0.0080) | (0.0008, 0.0078) |

Note: figures in the parentheses are standard deviations. Source: authors' estimation.

## 4. Discussion

In this study, the empirical results found that all explanatory variables, including China's cassava import volume from Thailand, China's cassava chips import price from Thailand, China's cassava starch import price from Thailand, the substitute crop prices of maize and wheat, and Thailand's cassava product export volume were statistically significant and caused increased volatility in the price of cassava. Few studies have examined the volatility of cassava prices in Thailand, especially for factors that impact price fluctuation. This study differs from Headey and Fan (2008) and Treesilvattanakul (2016) in terms of the factors that considerably influence the volatility of cassava prices, as those studies only considered demand–supply factors. Although this study's methods and factor variables are different from those of previous studies, the results are similar in terms of the perspective of volatility caused by several interrelated factors. Various studies in Thailand have investigated the fluctuation of agriculture commodity prices. Moreover, studies of cassava are limited, despite it being an important agricultural commodity in Thailand which is exported around the world and Thailand is the world's second largest agricultural producer. Additionally, the results indicate that the factor of the substitute crop price of maize has the highest effect on the volatility of the cassava chip price when

compared with the other factors such as the volatility price of the wheat, similar to the study of present situation and future potential of cassava in China (Yinong, Xiong and Shuren), which shows that maize also plays important role in the market, especially in southern China and in some specific industries. Therefore, the price change of maize will influence the volatility of cassava prices.

The causes of the volatility of cassava prices are consistent with previous studies on the effects of government policies and macroeconomic factors on agricultural commodity price volatility. Those findings reflect how government policy affects the volatility of agricultural commodity prices, resulting in increased price volatility for three major gain commodities (corn, soybean, and wheat) (Yang et al. 2010) and reduced agricultural price volatility (Crain and Lee 1996). Meanwhile, macroeconomic factors affecting price volatility include money balances, real per-capita income, the real exchange rate, the real deficit-to-income ratio (Apergis and Rezitis 2011), the international price index of edible oil, and foreign exchange rates (Yeasin et al. 2020). These findings are crucial and useful in determining government policy for agricultural commodity prices.

## 5. Conclusions and Policy Recommendation

Thailand is a significant global exporter of cassava, in which cassava chips are the main export products. The expansion of Thailand's cassava industry in the past ten years has resulted of the expansion of the export markets, especially the Chinese market. Considering the price volatility of cassava products in Thailand between 2000 and 2020, cassava chips clearly have significant price volatility, adapting to changes in export volumes during that period. This study examined the price volatility of cassava chips in Thailand from 2010 to 2020 using Bayesian GARCH-X method. Through the results of the estimation of GARCH-X (1,1) using Bayesian inference, we applied Bayesian estimation to GARCH-X model to obtain the posterior of the PDF condition, involving the computation of expected values for the properties of the GARCH-X model. Additionally, the estimation of parameters was adopted through Bayesian estimation, which assumes that parameters are random variables and have a probability distribution. This resulted in more accurate results where an estimator is closer to the true value, in particular where volatility results are affected by a factor that is difficult to predict. Bayesian estimation is an excellent tool to estimate the parameters of the model and examine the uncertainty of estimation for appropriate effects and accurate analysis.

The main conclusions of this study are that the volume of export cassava to China, the export price of cassava to China, China's cassava starch import price from Thailand, the price of maize, and the total export volume affect the volatility of cassava chip prices in the same direction. An increase in these factors led to an increased change in cassava chip price volatility. However, it was found that the price of wheat impacts cassava chip prices in a different direction, in which price increases of wheat cause a decrease of the change in cassava chip price volatility.

The implications of the findings are critical for the future policy of the Thai government regarding its main agricultural commodity price. This is because Thailand's agricultural commodity policy in the past has always focused only on domestic factors, especially product demand and supply. The same approach has been applied to cassava policy, in that is focused on factors that determine demand (increasing demand for cassava in various forms) and factors affecting supply (increasing efficiency and reducing the cost of production). Government policies are aimed at the equilibrium price level in the market. However, what affects the long-term well-being of farmers does not depend solely on the market equilibrium price because the equilibrium price in the market at any moment may cause farmers to profit or experience loss. Nevertheless, the estimation findings using the Bayesian GARCH-X model reflect that the price volatility of cassava chips was directly affected (both positively and negatively) by both domestic environmental factors (substitute crop price: maize, substitute crop price: wheat, and Thailand's cassava product export volume) and foreign environmental factors (China's cassava import volume from Thailand,

China's cassava chip import price from Thailand, and China's cassava starch import price from Thailand). For domestic environmental factors, price changes of substitute crops such as maize and wheat, which are important raw materials in Thailand for its feed industry, inevitably affect feed producers, changing the need for cassava, and eventually affected Thailand's cassava exports volume.

For foreign environmental factors that are significant factors in the change in cassava prices (export-focused goods), the factors of trading partners like China affect the change of price volatility of cassava chips. China is a country that frequently changes agricultural policy, such as in 2014 when the Chinese government supported ethanol production to develop biofuels and reduce dependence on oil imports. In 2018, China had a policy to support maize instead of cassava in the ethanol production industry, which resulted in a feed industry decline. These changes in China's policies affected a change in demand for cassava in China, which eventually affected demand for cassava imports from Thailand.

Volatility and changes in volatility for cassava prices also affect the long-term income and livelihoods of cassava farmers. In other words, increased price volatility results in farmers' price predictions being misleading. This effect affects both future supply and demand and also results in welfare losses for producers and consumers (Apergis and Rezitis 2011). Therefore, appropriate government cassava policy formulation should consider volatility and factors affecting price volatility. This reflects a particularly appropriate approach to agricultural policy, especially for food and energy agricultural commodities such as cassava (export-focused commodity). Moreover, the government should also consider changes to Thailand's trading partners. If policymakers signal changes to the factors that affect the price volatility of cassava chips (all studied factors, except the price of wheat), they should increase intervention measures or support farmers' production (e.g., production factors, replacement crops, and production credit), marketing (marketing credit), and poverty due to rising cassava chip price volatility levels. The findings of the present study confirm that the government policy formulation for every agricultural commodity, especially export-focused commodities, must consider the macro-environmental factors of the country and its key trading partners, especially when these environmental factors signal changes.

**Author Contributions:** Conceptualization, Y.C.; data curation, J.S.; methodology, J.S. and J.S.; writing—original draft, J.S.; writing—review & editing, B.L. All authors have read and agreed to the published version of the manuscript.

**Funding:** Chiang Mai University and Department of Agricultural Economy and Development, Faculty of Agriculture, Chiang Mai University, Grant number: 09/2021.

**Institutional Review Board Statement:** Not applicable.

**Informed Consent Statement:** Not applicable.

**Data Availability Statement:** The data used in this paper are available from the CEIC (https://insights.ceicdata.com) on 30 September 2020.

**Conflicts of Interest:** The authors declare no conflict of interest.

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
