# Peer review of "Modeling the Price Volatility of Cassava Chips in Thailand: Evidence from Bayesian GARCH-X Estimates"

_economies, doi:10.3390/economies9030132_

Round 1
Reviewer 1 Report
My comments are the following:
- The start of the introduction part is not cited at all. It should be supported by the relevant references. Actually, the whole Introduction part is not supported by references. The own thinking authors should leave for the conclusion part.
- I also found the Introduction part rather too long.
- The part LITERATURE REVIEW should be the part of the Introduction part. The authors are not writing the review paper.
- The discussion part is very small and it is also not supported by references.
- The conclusion part is missing.
Reviewer 2 Report
Main Comment
The article deals with Thailand is a significant exporter of cassava globally in which cassava chips are the main export products. Moreover, China is the most important export market for Thailand from 2000 until 2020. However, Thailand confronted the fluctuation of the cassava product price, cassava chips were a product with significant price volatility, adapting to changes in export volumes during that period. This study aims to analyze the volatility of cassava chip’s price in Thailand from 2010 to 2020. The data were collected monthly from 2010 to 2020, including cassava chips price of Thailand (Y), China’s cassava import volume from Thailand (X1), China’s cassava chips import price from Thailand (X2), China’s cassava starch import price from Thailand (X3), substitute crops price: maize (X4), substitute crops price: wheat (X5), Thailand’s cassava products export volume (X6). The volatility and the factor affecting the volatility in cassava chip’s price volatility were calculated using Bayesian GARCH-X. The results indicate that the increase in X1, X2, X3, X4 and X6 leads to increasing the change cassava chip price volatility. On the other hand, if substitute crops price: wheat (X5) increase, then cause the decreasing of the change of the volatility of cassava chip price. Therefore, appropriate government cassava policy formulated should take volatility and factors affecting price volatility into account. Additionally, government policy formulation of agriculture needs to consider macro-environmental factors of the country and its key trading partners, especially when these environmental factors signal changes.
The paper is well structured and informative, but authors are encouraged to go to another phase of manuscript review to address the following issues. The manuscript should be revised to address errors in several places. Several examples from the text follow (p - page):
(p.5) It is necessary to add a source to describe the Table 3.
(p.8) It is necessary to add a source to describe the Table 5.
(p.9) It is necessary to add a source to describe the Table 6.
(p.9) It is necessary to add a source to describe the Table 7.
(p.9-10) In conclusions, in addition to the above, I would expect an expression of opinion and perspective of the authors on the issue of the paper. Finally, the authors of the article should pay more attention to the overall writing and clarity of their article, which should support the demonstration of their findings.
References must be prepared in accordance with the journal template, I recommend preparing the references with a bibliography software package, such as EndNote, ReferenceManager or Zotero to avoid typing mistakes and duplicated references. In the text, reference numbers should be placed in square brackets [ ] and placed before the punctuation; for example [1], [1–3] or [1,3]. For embedded citations in the text with pagination, use both parentheses and brackets to indicate the reference number and page numbers; for example [5] (p. 10), or [6] (pp. 101–105).

Round 2
Reviewer 1 Report
The discussion part is still very small and authors changed the big part of the manuscript to the conclusion part.
The authors have to focus more on their results and discuss their results.
Author Response
Thank you for your suggestion, we added one more section for the discussion part to focus on the our results.
Reviewer 2 Report
The revised version of the manuscript is more extensive and of better quality.
The manuscript has been sufficiently improved to warrant publication in Economies.

Author Response
Thank you very much for your valuable recommendations.
Round 3
Reviewer 1 Report
The manuscript can be accepted.